# Resolution of Chronic Inflammation, Restoration of Epigenetic Disturbances and Correction of Dysbiosis as an Adjunctive Approach to the Treatment of Atopic Dermatitis

**DOI:** 10.3390/cells13221899

**Published:** 2024-11-18

**Authors:** Gregory Livshits, Alexander Kalinkovich

**Affiliations:** 1Department of Morphological Sciences, Adelson School of Medicine, Ariel University, Ariel 4077625, Israel; 2Department of Anatomy and Anthropology, Faculty of Medical and Health Sciences, Tel-Aviv University, Tel Aviv 6927846, Israel; kalinkovich@tauex.tau.ac.il

**Keywords:** skin, itch, specialized pro-resolution mediators, cytokines, keratinocytes

## Abstract

Atopic dermatitis (AD) is a chronic inflammatory skin disease with multifactorial and unclear pathogenesis. Its development is characterized by two key elements: epigenetic dysregulation of molecular pathways involved in AD pathogenesis and disrupted skin and gut microbiota (dysbiosis) that jointly trigger and maintain chronic inflammation, a core AD characteristic. Current data suggest that failed inflammation resolution is the main pathogenic mechanism underlying AD development. Inflammation resolution is provided by specialized pro-resolving mediators (SPMs) derived from dietary polyunsaturated fatty acids acting through cognate receptors. SPM levels are reduced in AD patients. Administration of SPMs or their stable, small-molecule mimetics and receptor agonists, as well as supplementation with probiotics/prebiotics, demonstrate beneficial effects in AD animal models. Epidrugs, compounds capable of restoring disrupted epigenetic mechanisms associated with the disease, improve impaired skin barrier function in AD models. Based on these findings, we propose a novel, multilevel AD treatment strategy aimed at resolving chronic inflammation by application of SPM mimetics and receptor agonists, probiotics/prebiotics, and epi-drugs. This approach can be used in conjunction with current AD therapy, resulting in AD alleviation.

## 1. Overview

Atopic dermatitis (AD) is the most common chronic inflammatory skin disease, affecting up to 20% of children and 10% of adults, depending on ethnicity [1,2,3]. It is a complex and heterogeneous immune-mediated skin condition characterized by distinct phenotypes, each exhibiting various molecular variants [4,5,6,7]. Noxious environmental stimuli, immune dysregulation, genetic and epigenetic factors, impaired epidermal barrier integrity, and skin and gut microbiota abnormalities (dysbiosis) are all considered to play an important role in initiating and maintaining the state of chronic inflammation in AD and contribute to the control of the disease phenotype [8,9].

Being an immune-mediated disease, AD requires the use of immune-targeted approaches for its treatment. AD emerged as the T helper 2 (Th2) disease, and this is supported by multiple observations, including increased levels of cytokines and chemokines secreted by Th2 cells, such as interleukin 4 (IL-4), IL-5, IL-13, CCL17, CCL18, and CCL22 [10,11,12]. AD is also characterized by the Th22-skewed phenotype with the overproduction of IL-22, while the contributions of the Th1 and Th17 axes vary depending on the AD endophenotype [13,14].

These observations lead to the development of targeted therapies, in particular, biological drugs (mainly monoclonal antibodies to cytokines and cytokine receptors) and Janus kinase inhibitors (JAKi) [15,16,17]. Although these treatments greatly improve AD management, they are not without serious side effects, mainly due to non-specific immune suppression [18,19,20]. Moreover, unlike psoriasis, where cure rates reach 80%, targeted cytokine inhibition in patients with AD has not demonstrated comparable efficacy [17]. This discrepancy may be due to the heterogeneous nature of AD, which includes a wide range of phenotypes and endotypes [21,22]. However, persistent inflammation remains a constant characteristic of any type of chronic AD [23,24,25]. The use of monoclonal antibodies against key cytokines and their receptors involved in AD-associated inflammation, such as IL-4 and IL-13, has shown therapeutic effects in severe AD [17,24,26].

However, as in the case of JAKi, they are not able to resolve chronic inflammation and also have serious side effects due to their nonspecific immunosuppressive activity [27,28,29]. In this regard, an application of the agents capable of resolving chronic inflammation may be an attractive therapeutic approach for AD. Inflammation resolution is an active process involving the activation of signaling pathways during inflammation initiation with the subsequent generation of specialized pro-resolution mediators (SPMs) derived from polyunsaturated fatty acids (PUFAs) that activate resolution mechanisms [30]. These mediators include resolvins, lipoxins, protectins, and maresins acting through a range of G protein-coupled receptors (GPCRs) [31]. Notably, PUFA derivative levels are decreased in patients with AD, and PUFA supplementation ameliorates the disease manifestations [32,33,34,35]. Moreover, recent Mendelian randomization analysis suggests that elevated levels of n-3 PUFA and docosahexaenoic acids (DHA), which is a derivative of n-3 PUFA, are associated with a lower risk of AD [36]. Several preclinical studies in AD models revealed clear beneficial effects of SPMs [37,38], supporting the validity of the idea of using agents that resolve chronic inflammation to treat AD.

The prevalence of AD has tripled in the past 30 years in industrial countries [39], raising the possibility that environmental factors, acting through epigenetic mechanisms, may be important. Indeed, epigenetic alterations (DNA methylation, histone modifications, and non-coding RNA-mediated regulation) were shown to contribute to key pathogenic events, such as immune activation, T-cell polarization and keratinocyte dysfunction in AD [40,41]. This suggests an application of drugs selectively influencing the epigenetic landscape (epidrugs) in AD. In support, some studies report on the efficacy of micro-RNA (miRNA) targeting in AD murine models [42,43,44]. The epigenetic regulators have been shown to modulate chromatin structure in epidermal keratinocytes by influencing the function of genes, mainly epidermal differentiation complex (EDC) genes, which are associated with proliferation and differentiation processes in keratinocytes and their precursors [45]. Thus, we can propose the epidrugs that are most suitable for inhibiting inflammation (and/or promoting its resolution) as well as supporting keratinocyte function as potential AD therapeutic agents.

AD is associated with marked changes in the skin microbiota composition (dysbiosis), specifically loss of commensal diversity and dominant colonization by pathogenic *Staphylococcus* bacteria, primarily *S. aureus* [46]. This dysbiosis is associated with more severe disease, more pronounced type 2 immune abnormality, and barrier impairment [47]. Conversely, restoration of commensal diversity may precede and indicate disease resolution [48,49,50]. The development of AD is also characterized by gut dysbiosis, where, as in the skin, the abundance and diversity of commensals are reduced while the abundance of pathobionts is increased [51,52]. AD-associated gut dysbiosis results in impaired intestinal barrier integrity (leaky gut syndrome) and reduced production of short-chain fatty acids (SCFA), mainly the anti-inflammatory butyrate and propionate. Notably, the absence of SCFA-producing microbes preceded AD symptoms [46] and was found to be positively correlated with disease severity in infants [53]. Moreover, butyrate has been shown to attenuate skin inflammation exacerbated by *S. aureus* [54], thereby supporting the existence of a gut-skin axis playing a detrimental role in the pathogenesis of AD [55,56].

Based on these findings, we suggest that the combination of SPM mimetics and receptor agonists with dysbiosis and epigenetic disturbance correctors has translational potential to alleviate the pathological manifestations of AD. The ultimate goal of this strategy is to effectively resolve chronic inflammation. Evidence supporting this new AD treatment approach prompted the present review.

## 2. The Major Role of Chronic Inflammation in the Pathogenesis of AD

Despite careful, long-term study [4,8,57], the mechanisms underlying the pathogenesis of AD remain unclear. However, recent research has shown notable progress. Major advances are related to a better understanding of genetic and epigenetic regulation [39,40,58], the role of innate immunity [59,60], the involvement of T-cell subsets and cytokines [61,62,63], the role of skin and gut dysbiosis [49,50,64], and mechanisms of epithelial barrier disruption in the pathogenesis of AD [65,66,67,68]. It is assumed that all these mechanisms underlie the development and maintenance of chronic inflammation, which is considered a key pathogenic factor in AD [61,69,70].

Over the last 3 decades, the prevailing hypothesis has been that acute AD is predominantly driven by Th2 cytokines, with a shift to a Th1-dominant milieu in the chronic phase [71,72,73]. However, a detailed study of the transcriptomic changes that characterize the transition from acute to chronic lesional skin has led to a paradigm shift in the immunopathology of AD [11,74]. Of particular significance are the results of the study by Tsoi et al. [74], in which RNA sequencing was used to characterize non-lesional, acute and chronic lesional skin from patients with AD and healthy skin. Genetic transcriptome analysis revealed that 74% of genes whose regulation was disrupted in acute AD were also observed in chronic AD, including genes involved in epithelial differentiation (IL-20, KRT16, KRT6B, S100A8, and S100A9), antimicrobial and immunomodulatory chemokines (CXCL1, CXCL6), T cell regulation (TNIP3, CLEC3A) and Th2 cell differentiation (IL-19). In contrast, only 34% of the expressed genes found in chronic skin lesions were also altered in acute lesions [74]. Moreover, acute skin lesions and cytokine profiles often exhibit a biphasic profile, indicating an ongoing transition to a chronic phase [74]. These findings indicate that acute lesions result in widespread activation of the immune system, while chronic lesions leave these genes intact.

Specifically, genes related to most Th-cell subsets, including Th1-, Th2-, Th17-, and Th22-related genes, were found to be dysregulated already in acute lesions, and these genes remained or are further dysregulated in chronic lesions [74]. In contrast to the paradigm suggesting a Th2 to Th1 switch, this study reported no association of the immunological switch with the transition from acute to chronic stage, but rather, progressive immune activation of all Th subsets was observed. 

Another important observation reported in this study is that a relatively small number of genes are differentially regulated in chronic versus acute AD: 29 genes showed increased and 13 decreased expression levels, suggesting relatively small differences between acute and chronic lesions. Altogether, these findings suggest that the transition of acute-to-chronic AD is associated with quantitative rather than qualitative changes in cytokine responses.

In addition to the above-mentioned T-cell subsets, other immune cells, as well as keratinocytes, are deeply involved in skin inflammation, in which the coexistence of all these cells creates a complex, self-sustaining vicious cycle that underlies AD pathogenesis. As demonstrated in Figure 1, activated keratinocytes produce numerous chemokines, including thymus- and activation-regulated chemokine (TARC)/CCL17, regulated on activation, normal T-cell-expressed and secreted (RANTES)/CCL5, monocyte chemotactic protein (MCP)-4/CCL13, eotaxin-3/CCL26, and some others, which attract and activate Langerhans cells (LCs), dendritic cells (DCs), eosinophils, basophils, mast cells, type-2 innate lymphoid cells (ILC2), and Th17 cells. Keratinocytes also produce thymic stromal lymphopoietin (TSLP), which, in concert with all mentioned mediators, stimulates type 2 innate lymphoid (ILC2) cells to produce IL-5 and IL-13 and also Th2 cells to produce IL-1β, IL-4, IL-6, IL-8, IL-13, IL-18, IL-25, IL-31, and IL-33. In particular, IL-4 and IL-13 recruit eosinophils and stimulate B cells for the production of IgE and several chemokines, such as CCL5, CCL11, CCL17, CCL18, and CCL-22.

IL-31 stimulates keratinocytes, whereas IL-33 decreases the expression of filaggrin (FLG) by keratinocytes, a major epidermal protein, thereby promoting itch (pruritus) and skin barrier disruption. In addition, IL-4 and IL-13 suppress the production of antimicrobial peptides (AMPs), weakening the barrier function against microbes. IL-4, IL-5, and IL-13 further amplify the Th2 immune responses and exacerbate AD. Th17 and Th22 cells also increase in number, leading to enhanced secretion of pro-inflammatory mediators. In addition to the Th2 immune response, IL-17 from Th17 cells and IL-22 from Th22 cells induce acanthosis in the epidermis, which contributes to the formation of intractable lichenified skin lesions. Produced by activated DCs, IL-12, together with IL-18, activate Th1 cells to produce IFNγ, which, in turn, stimulates keratinocytes. This pro-inflammatory vicious cycle drives and maintains AD-associated chronic skin inflammation, thereby exacerbating the disease manifestations [61,62,75,76,77,78,79].

## 3. Epigenetic Profile of AD

Epigenetic changes (i.e., modifications of genetic expression information not associated with changes in DNA sequence) play an important role in normal development, cell-specific gene expression, and the pathogenesis of many complex diseases [80,81], including inflammatory skin disorders [4,82,83,84,85]. The major molecular epigenetic mechanisms include DNA methylation, histone modification, and regulation by non-coding RNAs (ncRNAs), mainly by miRNAs [86]. The key players governing these epigenetic modifications include three families of epigenetic enzymes, namely writers, readers, and erasers.

Writers are a group of enzymes that act on histones and add small covalent modifications such as methyl and acetyl groups. DNA methyltransferases (DNMT1, DNMT3A, and DNMT3B) add methyl groups to the fifth carbon of cytosine residues that are linked by a phosphate to a guanine nucleotide (i.e., a CpG dinucleotide). Most of the CpG dinucleotides, which are distributed irregularly across the genome, are methylated. The exception is CpG dinucleotides within CpG islands that are usually unmethylated [87,88]. Histone acetyltransferases (HATs) add acetyl groups, and histone methyltransferases (HMTs) add methyl groups to histones.

Readers are a diverse group of proteins, such as the bromodomain and extra-terminal domain (BET) family of proteins. They identify histone modifications made by writers and mediate downstream biological events. Erasers remove epigenetic modifications. Histone demethylases (HDMs) remove methyl groups, and histone deacetylases (HDACs) remove acetyl groups from histones. In humans, 18 different mammalian HDACs have been identified, of which class III HDACs (Sirtuins, SIRTs) are involved in the development of asthma [89] and AD [90]. All these epigenetic modifications play a critical role in altering chromatin conformation, leading to either transcriptional repression or gene activation [91,92,93].

Significant differences in DNA methylation patterns were observed between lesional and intact epidermis in AD patients, suggesting that aberrant DNA methylation may play a role in the pathogenesis of AD [94]. In this study, substantial hypermethylation changes in the S100A proteins OAS2 and KRT6A, involved in the regulation of innate immunity [95], have been found to be related to their increased expression in lesional skin. In another study [96], global hypomethylation was observed in DCs and monocytes from AD patients compared to healthy controls along with locus-specific hypomethylation at the high-affinity receptor for IgE (FCER1G) promoter in correlation with its overexpression. Increased DNA methylation of the FLG gene, a major structural protein in the stratum corneum, was observed in the lesional epidermis of patients with severe AD compared to the non-lesional epidermis [97] and was also associated with an increased AD risk [98]. Interestingly, although the CD4^+^ T cells from AD patients do not show significant changes in global methylation patterns compared to healthy controls [99], CD4^+^ T cells expressing the cutaneous lymphocyte antigen (CLA) (CD4^+^CLA^+^ T cells) showed significant differences in DNA methylation in 40 genes, including IL-13 gene, compared with healthy controls [100]. These results suggest the involvement of epigenetic regulation in the functional activity of T cells homing to the skin in AD pathogenesis.

DMT1 mRNA expression level is significantly lower in peripheral blood mononuclear cells (PBMCs) from AD patients compared with healthy controls [101]. mRNA and protein expression of TSLP is upregulated in the skin lesions from patients with AD in association with promoter hypomethylation (detected by using bisulfite sequencing), suggesting that DNA demethylation (detected by using 5-aza, a DNA methyltransferase inhibitor) of a regulatory region of TSLP may contribute to its overexpression in AD skin lesions [102]. Bisulfite pyrosequencing of the promoter region of human AMP beta-defensin-1 (HBD-1) showed significantly higher methylation frequencies at the CpG 3 and CPG 4 sites in AD lesional samples than in non-lesional AD skin and healthy skin samples [103], suggesting that promoter DNA methylation contributes to HBD-1 deficiency in AD. Additionally, highly methylated SNPs in the gene encoding kinesin family member 3A protein (KIF3A) were found in the skin samples from AD patients compared with healthy controls [104]. In this study, KIF3A knockout mice demonstrated impaired junctional proteins and increased susceptibility to developing AD, suggesting an important role of KIF3A in maintaining skin barrier integrity.

Currently, only a few studies have focused on AD-specific histone modifications. They have found, for example, that the expression of the SIRT1 gene, involved in several cellular pathways and associated with the beneficial effects on skin aging [105], is down-regulated in the lesions of AD patients compared to normal individuals [90]. In a study of human and mouse keratinocytes, it was shown that physical interaction between HDAC1 and the FLG promoter [detected by chromatin immunoprecipitation (ChIP) assay] stimulates FLG promoter activity, which was significantly suppressed by TNFα and IFNγ [106].

The vast majority of the results linking epigenetics and AD have been obtained from the study of ncRNAs, primarily miRNAs [107,108,109]. As illustrated in Figure 2, miRNAs are involved in the regulation of key mechanisms in AD pathogenesis, such as T-cell imbalance, chronic inflammation, and skin barrier dysfunction. miR-155 is one of the most up-regulated miRNAs in AD patients [110], and its plasma levels positively correlate with the disease severity, the percentage of Th17 cells and IL-17 expression [111]. In this study, miR-155 was found to be predominantly expressed in activated skin infiltrating immune cells, presumably due to reduced expression of CTLA-4, an immune checkpoint receptor that has an inhibitory effect on T-cell responses, thereby promoting chronic skin inflammation. In addition, miR-155 stimulates the differentiation of T regulatory cells (Tregs) and Th17 cells by targeting suppressors of cytokine signaling 1 (SOCS1), the important negative regulator of JAK/signal transducer and activator of transcription (JAK/STAT) signaling pathway [112]. In a murine model of AD, silencing of miR-155 attenuated epidermal thickening, reduced inflammatory cell infiltration and Th2 cytokine secretion as well as increased the expression of protein kinase inhibitor α (PKIα) and tight junction proteins; reduced expression of TSLP and IL-33 was also found in these mice [113].

An increased expression of miR-146a in keratinocytes of AD patients appears to play an anti-inflammatory role. Indeed, miR-146a inhibits the expression of NF-κB upstream elements, such as caspase recruitment domain-containing protein 10 (CARD10), tumor necrosis factor receptor-associated factor 6 (TRAF6), and interleukin-1 receptor-associated kinase 1 (IRAK1), thereby reducing IFNγ- and NF-κB-activated chronic skin inflammation [114]. Elevated miRNA-146a expression in human primary keratinocytes stimulated with IFNγ, TNFα, or IL-1β is associated with reduced expression of numerous pro-inflammatory factors, including CCL5 and CCL8 [115]. In a murine model of AD, miR-146a-deficient mice developed skin inflammation, characterized by increased accumulation of infiltrating immune cells and increased expression of CCL5 and CCL8 in the skin [115].

Another example is miR-124. It inhibits the expression of p65, a member of the NF-κB family involved in the inflammatory reactions. The decreased expression of miR-124 was observed in lesional skin samples from chronic AD [116]. In this study, keratinocyte stimulation with IFNγ or TNFα resulted in upregulation of IL-8, CCL5 and CCL8 partially reversed by miR-124. These data indicate that miR-124 suppresses NF-κB-dependent inflammatory responses in keratinocytes and chronic skin inflammation in AD, suggesting elevation of miR-146a as a potential therapeutic approach in AD.

Increased expression of several miRs has been suggested to be associated with the worsening of AD, suggesting that their inhibition may result in attenuation of AD manifestations. For example, the viability of human keratinocytes and release of IL-1β and IL-6 after stimulation with TNFα and IFNγ are negatively associated with the expression of miR-375-3p through the activation of Yes-associated protein 1 (YAP1) and lymphoepithelial Kazal type inhibitor (LEKTI) inflammatory pathways [117]. The negative role of miR-375-3p in AD pathogenesis is shown in a clinical study in which the severity of AD is positively associated with miR-375-3p levels in saliva samples [118]. Increased expression of miR-10a-5p in keratinocytes of AD patients compared with healthy controls is associated with reduced keratinocyte proliferation and migration through targeting hyaluronan synthase (HAS3), a damage-associated positive regulator of keratinocyte function, thereby potentially impairing skin barrier in AD [119]. Upregulation of miR-29b in lesional skin and serum samples from AD patients is associated with enhanced keratinocyte apoptosis through inhibiting Bcl-2-like protein 2 (Bcl2L2), contributing to epithelial barrier dysfunction impaired in AD [120].

In contrast, elevated plasma miR-151a levels in AD patients compared with healthy controls appear to be beneficial as they are associated with downregulation of IL-12 receptor β2 (IL12RB2) expression, a subunit of the IL-12 receptor [121]. Therefore, downregulation of IL12RB2 expression by miR-151a may reduce the responsiveness of immune cells to IL-12 signaling. The binding of this receptor activates the JAK/STAT pathway involved in the production of IFNγ and TNFα by T cells and natural killer (NK) cells, thereby enhancing inflammation [122].

Stimulation of human epidermal keratinocytes with IL-13 results in the reduced expression of miR-143, whereas overexpression of miR-143 mimics IL-13-induced downregulation of FLG, loricrin, and involucrin in epidermal keratinocytes via targeting IL-13Rα1 [123]. These data suggest that activation of miR-143 may attenuate AD and that this miRNA may serve as a potential preventive and therapeutic target in AD. In AD lesional skin, miR-335 expression is downregulated, whereas SOX6 is upregulated throughout the epidermis, resulting in reduced keratinocyte differentiation and impaired skin barrier [124].

Bioinformatics analysis of microarray data identified two downregulated differently expressed miRNAs (DEMs), namely, let-7a-5p and miR-26a-5p [125]. let-7a-5p potentially targets the chemokine receptor CCR7, overexpressed in T cells and DCs in AD lesions [126]. miR-26a-5p probably regulates the HAS3 gene, involved in the synthesis of hyaluronic acid, a major component of the extracellular matrix and overexpressed in AD lesions compared with healthy skin and non-lesional AD skin [127]. miR-939 is found to be highly upregulated in *S. aureus*-stimulated keratinocytes and AD lesions, accompanied by increased expression of several matrix metalloproteinases (MMPs) to promote the colonization of *S. aureus* and exacerbated *S. aureus*-induced AD-like skin inflammation [128]. 

In addition to miRNAs, long non-coding RNAs (lncRNAs) are also likely to be involved in the pathogenesis of AD. In the murine model of AD, the expression of lncRNA MALAT1 is upregulated in the skin, and its knockdown represses NLRP3 inflammasome activation and mitigates Th1/Th2 imbalance, thus potentially ameliorating AD [129]. In another study, downregulation of MALAT1 expression resulted in the attenuation of the expression of the CCR7 gene via a competing endogenous RNA mechanism involving miR-590-5p [130]. This pathway effectively inhibits TNFα/IFNγ-induced keratinocyte proliferation and inflammation. A study of macro-phages revealed that lncRNA, lncFAO, contributes to inflammation resolution and tissue repair in mice by promoting fatty acid oxidation in macrophages [131].

Taken together, these findings indicate the profound involvement of ncRNAs, especially miRNAs, in AD pathogenesis, suggesting their targeting as a promising therapeutic approach for AD. In support, one of the new anticancer drugs, the pan-HDAC inhibitor belinostat, has been shown to promote keratinocyte differentiation and restoration of skin barrier function by upregulating the expression of miR-335, which, in turn, targets the transcription factor SOX6 to promote terminal differentiation of keratinocytes [124].

## 4. The Role of Skin-Gut Axis in the Pathogenesis of AD

Healthy skin is colonized by a variety of commensal microorganisms while preventing the growth and penetration of pathogenic microorganisms [132,133]. AD is associated with dramatic changes in the skin microbiota, particularly the loss of commensal diversity and dominant colonization by pathogenic *Staphylococcus* bacteria, mainly *S. aureus* [46] and also *S. capitis*, and *S. lugdunensis* [134]. Not all patients with AD are colonized with *S. aureus*; however, a high abundance of *S. aureus* in AD is associated with more severe disease and greater type 2 immune deviation, allergen sensitization, and barrier disruption than in non-infected patients [135,136]. Analysis of *S. aureus* clinical isolates demonstrates that most pathogenic strains predominate in AD skin in correlation with disease severity [50,137], whereas restoration of commensal assortment has been suggested to precede and predict disease resolution [138,139].

Commensal skin microbiota produces tryptophan metabolites, such as indole-3-aldehyde (IAld), that block Th2 induction through the aryl hydrocarbon receptor (AHR) expressed mainly by keratinocytes [140]. The skin of AD patients displays a lower level of IA1d compared to that of healthy subjects. Collectively, these findings indicate that skin dysbiosis plays a detrimental role in AD pathogenesis, mainly via exacerbating the Th2-associated immune response. Notably, this immune dysregulation can influence the skin microbiota in AD. For example, *S. aureus* binds more efficiently to IL-4-stimulated skin samples compared to untreated control skin in a murine model of AD, suggesting that a Th2 inflammatory environment promotes skin binding by *S. aureus* [141]. Additionally, in cultured human keratinocytes, Th2 cytokines inhibit the expression of AMPs, namely human β-defensin 2 (HBD2), HBD3 and cathelicidin (LL-37), which have anti-staphylococcal activity [142,143]. A similar reduction in AMP expression was observed in skin explants from AD patients compared to healthy controls, which is reversed by neutralizing antibodies against IL-4, IL-10, and IL-13 [144,145]. Together, these data suggest the existence of a bidirectional link between skin dysbiosis and chronic inflammation in AD, and the mechanisms involved in the process are schematically summarized in Figure 3.

AD pathogenesis is also characterized by gut dysbiosis in which, similar to skin, the prevalence and diversity of commensals are reduced, whereas the prevalence of pathobionts is increased [52,55,146,147]. AD-associated gut dysbiosis has two major detrimental consequences: compromised gut barrier integrity (leaky gut) and reduced production of short-chain fatty acids (SCFAs), mainly anti-inflammatory butyrate and propionate. A leaky gut is followed by increased blood levels of intestinal endotoxins such as lipopolysaccharide (LPS), which, through binding toll-like receptor 4 (TLR4) on macrophages and T cells, trigger signaling cascades that culminate in the production of pro-inflammatory cytokines (TNFα, IL-1β, IL-6, IL-12) and type I interferons participating in the creation of chronic inflammation [148,149]. The main producers of butyrate and propionate are *Faecalibacterium prausnitzii* [150,151] and *Dialister* [152]. In fecal samples from AD patients, their numbers were reduced compared to healthy controls [153,154]. Moreover, a recent Mendelian randomization analysis suggests an inverse correlation between *Dialister* and AD [155]. Lack of SCFA-producing microbes preceded AD symptoms [46] and is found to be positively correlated with the disease severity in infants [53], suggesting that SCFAs play a role in improving AD manifestations [156]. In support, high fecal levels of propionate and butyrate were found to be associated with decreased atopic sensitization in children [157]. Moreover, butyrate administration attenuated skin bleeding, scarring, dryness, abrasions and erosions in an AD murine model [158]. Notably, butyrate increases the expression of mRNA and protein levels of FLG in normal human epidermal keratinocytes [159] and attenuates *S. aureus*-aggravated skin inflammation with decreased IL-13, IL-33, and leukocyte infiltration in the skin [54]. Moreover, butyrate derivative BA-NH-NH-BA reduces skin colonization by *S. aureus* and ameliorates *S. aureus*-induced production of IL-6 in a murine model of AD [160]. Overall, these results suggest that gut dysbiosis-induced SCFA deficiency is involved in AD-associated *S. aureus* accumulation in the skin, keratinocyte activation, and skin inflammation, thereby reinforcing the deleterious role of the skin-gut axis in AD pathogenesis (Figure 3) [52,147,161]. Remarkably, there is evidence suggesting a link between epigenetic disturbance and skin dysbiosis in AD pathogenesis. Indeed, low levels of propionate on the skin surface of AD patients are associated with increased expression of HDAC2 and HDAC3 [162]. Reduced concentrations of butyrate are associated with increased growth of *S. aureus* via inhibition of acetylation of histone H3 lysine 9 (AcH3K9) in human AD keratinocytes [160].

## 5. Failed Resolution of Chronic Inflammation as a Key Mechanism of AD Pathogenesis

The abundant data unequivocally show that chronic skin inflammation is the main pathogenetic mechanism of AD. Its successful resolution can, therefore, lead to a weakening of disease manifestations. Inflammation resolution is a process primarily driven by dietary polyunsaturated fatty acid (PUFA)-derived specialized pro-resolving mediators (SPMs) through the sequential enzymatic activities of several lipoxygenases and hydrolases. The omega-3 PUFA docosahexaenoic acid (DHA) serves as the substrate for D-series resolvins (RvD1-RvD6), maresins (MaR1, MaR2, and eMaR), and protectins/neuroprotectins (PD1/NPD1), cysteinyl-SPMs (MCTR1-R3, PCTR1-R3, and RCTR1-R3) as well as n-3 docosapentaenoic acid (DPA)-derived SPMs (PD1n-3 DPA), while eicosapentaenoic acid (EPA) serves as a substrate for E-series resolvins (RvE1-RvE4). Lipoxins (LXs), such as LXA4 and LXB4, are derivatives of the omega-6 PUFA arachidonic acid (AA) [163]. SPMs exert their biological actions via cognate G protein-coupled receptors (GPCRs), namely ALX/FPR2, GPR32, ChemR23, BLT1, GPR18, GRP37, and LGR6. The underlying mechanisms of inflammation resolution, which are still under investigation, have been comprehensively reviewed (e.g., [164,165,166,167,168]). In short, this multistep process involves cessation of neutrophil infiltration, counter-regulation of pro-inflammatory cytokines and chemokines, reduction of reactive oxygen species (ROS) and NLRP3 inflammasome generation, induction of neutrophil apoptosis and their efferocytosis by macrophages, accumulation of anti-inflammatory M2 macrophages, and induction of Tregs, all of which initiate the healing processes and culminate in a return to tissue homeostasis. Uncontrolled, excessive inflammation or failure to promptly resolve inflammation results in non-resolving inflammation, which is often chronic or transient and recurrent. Reduced SPM production and functional activity are hypothesized to be the underlying mechanisms of failure to resolve inflammation [169]. Inflammation resolution has emerged as a critical physiological process that protects host tissues from prolonged or excessive inflammation that can become detrimentally chronic [168].

One of the most remarkable features of SPM receptors (in particular, ALX/FPR2) is their unusually high degree of molecular heterogeneity in recognizing a wide range of ligands, indicating that not only each receptor is engaged by different SPMs but also that a single SPM can act through the activation of different receptors [170]. This mutual redundancy is reflected in the relationships between the SPMs, their receptors and immune cells during inflammation resolution. Indeed, it has been repeatedly demonstrated that most SPMs bind various receptors expressed by neutrophils, macrophages, lymphocytes, and other immune cells, thereby ensuring effective inflammation resolution [171,172,173,174]. Accordingly, dysregulation of an SPM/receptor/immune cells cross-talk is suggested as a main cause of inflammation chronicity [31,175], which maintains and exacerbates multiple associated disorders [176,177,178,179]. Hence, enhancing this cross-talk by upregulating SPM functional activity and/or targeting SPM receptors may potentially be an effective approach to attenuate the manifestations of chronic inflammation-associated disorders, including AD.

Importantly, AD patients are characterized by reduced levels of the derivatives of omega-3 PUFAs, suggesting that the natural mechanisms of inflammation resolution in AD are suppressed. In accordance with this idea, transcriptomic and lipidomic pro-filing of patients with AD revealed decreased EPA and DHA levels and omega-3/omega-6 PUFA ratio in the serum and in lesional and even non-lesional skin [180]. Moreover, an increased ratio of pro-inflammatory vs pro-resolving lipid media-tors was found, overall suggesting a strong inflammatory background towards the maintenance of chronic inflammatory status in AD, displaying no tendency of its re-solving. Two recent Mendelian randomization studies suggested a negative correlation between AD risk and omega-3 PUFA [36] and DHA [181] serum levels. In the study by Lin et al. [36], leave-one-out analyses revealed that the protective effect of omega-3 PUFA was mainly mediated by SNP (rs174546) located in the fatty acid desaturase (FADS) gene cluster, highlighting its importance in the fatty acid synthesis pathway in the development of AD. In another recent study [181], DHA levels were negatively associated with the levels of the proinflammatory cytokine TNFSF14. Moreover, a positive correlation was found between TNFSF14 levels and the risk of AD. These results imply that genetic factors have a potential role in the observed PUFA deficiency in AD. 

The potential beneficial role of inflammation resolution in patients with AD was first observed in a 1987 study in which taking 10 g of fish oil (containing approximately 1.8 g of EHA) daily for 12 weeks was associated with reduced AD symptomatology [181]. Since then, similar effects of omega-3 PUFAs and their derivatives in AD patients have been repeatedly reported (e.g., [34,182,183,184,185]), although the results of meta-analyses are not consistent [186,187,188,189]. One reason for this discrepancy may be that in cases of oxidative stress, which is a key feature of chronic inflammation in AD [190], omega-3 PUFAs become very oxidized [191,192]. This, in turn, may lead to the formation of highly bioactive lipid peroxidation products such as 4-hydroxynonenal (4-HNE) and malondialdehyde (MDA), which are known to induce inflammation and other immune changes, including increased cytokine secretion and activation of inflammatory transcription factors [193,194]. These findings point to the existence of complex relationships between PUFA oxidation and their biological activity [195,196,197], which may influence the beneficial effects of omega-3 PUFA in patients with oxidative stress-related conditions, including AD. 

Several preclinical studies have shown clear preventive/therapeutic effects of SPMs in skin inflammation models. For example, in a 2,4-dinitrofluorobenzene (DNFB)—a murine model of AD-like skin lesion, intraperitoneal injection of RvE1 significantly decreased ear swelling and improved skin lesions, accompanied by decreased serum IgE levels and production of IFNγ and IL-4 by activated T cells, and reduced infiltration of eosinophils, mast cells and T cells in the skin lesions [37]. In a psoriatic murine model, RvE1 suppressed the inflammatory cell infiltration, epidermal hyperplasia and mRNA expression of IL-23 in the skin and inhibited migration of cutaneous DCs and γδ T cells, a major IL-17-producing cell population in mice, presumably via binding to the BLT1 receptor [198]. In a murine model of allergic inflammation induced by intraperitoneal injections of ovalbumin (OVA), RvE1 suppressed the production of IL-23 and IL-6 in the lung [199] as well as reduced airway eosinophil and lymphocyte recruitment, IL-13 secretion, and IgE levels [200,201]. In a murine model of imiquimod (IMQ)-induced psoriasiform skin inflammation, RvD1 downregulated skin mRNA expression of IL-17, IL-22, IL-23, and TNFα via inhibition of MAPKs and NF-κB signaling pathways [202]. In a similar model, protectin D1 (PD1) reduced skin thickness, redness, and scaling, decreased IL-1β, IL-6, IL-17, and CXCL1 mRNA expression in the skin and serum via inhibition of STAT1 and NF-κB signaling pathways [203]. Notably, decreased mRNA expression of IL-1β, IL-6, IL-8, and CCL17 was also observed in keratinocytes of PD1-treated mice. 

Since SPMs function through triggering their cognate receptors, these data suggest the expression of SPM receptors in keratinocytes. Indeed, human keratinocytes were shown to express ChemR23 [204] and BLT1 [205], which are both receptors for RvE1 and RvE2, as well as ALX/FRP2, the receptor for RvDs and LXA4 [206]. In the latter study, ALX/FRP2-deficient mice showed an endogenous defect in re-epithelialization, and topical application of RvD2 accelerated re-epithelialization during skin injury and enhanced migration of human epidermal keratinocytes in an ALX/FRP2-dependent manner. These data indicate that resolvins play a direct, keratinocyte-mediated role in the tissue repair program. Another study showed that mouse and human keratinocytes use FPR2 to detect *S. aureus* and trigger antimicrobial defenses in the skin [207]. Moreover, in the murine IMQ-induced psoriasiform inflammation model, topical application and systemic administration of RvD3 results in reduced skin inflammation, acute pain and itch [208].

Overall, these observations suggest that deficiency of SPMs in AD not only reduces the pro-resolving activity of immune cells but also affects the functional activity of SPM receptor-expressing keratinocytes, along with increasing the pain and itch associated with AD. In support, increased levels of omega-6-PUFA-derived major itch mediators, such as leukotriene B4 (LTB4), thromboxane B2 (TXB2), 12-hydroxyeicosatetraenoic acid (12-HETE), prostaglandin E2 (PGE2) and PGF2, as well as BLT1 (receptor for RvE1 and RvE2), have been detected in affected and non-affected skin of AD patients [180]. The authors suggest that in patients with AD, even healthy-looking skin is prone to a pro-inflammatory status and pro-itchy conditions, which are most likely caused by a systemic disorder [209].

Itch triggers scratching responses following activation of pruriceptors in primary sensory neurons, which are a subset of nociceptors and also express transient receptor potential vanilloid 1 and ankyrin 1 (TRPV1 and TRPA1, respectively) [210,211,212]. Both TRPV1 and TRPA1 are co-expressed in a wide range of sensory nerves, where they integrate multiple noxious stimuli, and in non-neuronal cells, such as keratinocytes, mast cells, DCs, and endothelial cells, serving as nociceptive sensors that amplify the inflammatory process [211]. AD is commonly associated with persistent itch [213], and SPMs have been shown to alleviate it by reducing skin inflammation. Indeed, topical application of LXA4 reduces the severity of childhood AD and improves the quality of life through controlling skin inflammation by downregulating TLR4, p-ERK1/2, and NF-κB signaling, and proinflammatory cytokines [214,215]. RvD3 administration prevents the development of psoriasiform itch and skin inflammation via inhibition of TRPV1 signaling in mouse and human dorsal root ganglion (DRG) neurons, which may account for the anti-itch effects of RvD3 [208]. Another study reported an important role of spinal glial cells in driving chronic itch [216]. Intrathecal administration of PD-1, PD1n-3 DPA, and the new analog 3-oxa-PD1n-3 DPA significantly reduced scratching for several hours [217]. These observations suggest the expression of SPM receptors on neuronal cells. In support, receptors for LXA4, RvD1, RvD3, RvD5 and some of their aspirin-triggered analogs were shown to bind to either astrocytic and neuronal ALX/FPR2 in both humans and mice or neuronal and microglial GPR32 receptor in humans. GPR18, the receptor for RvD2, is expressed in all brain cells except oligodendrocytes, and GPR37, the receptor for PD1, is expressed in all brain cells except microglia. RvE1/2 binds the ChemR23 receptor in astrocytes and the BLT1 receptor in astrocytes, microglia and neurons. MaR1 can bind and activate surface LgR6 receptors in astrocytes and neurons and intracellular RORα receptors expressed in all brain cells except oligodendrocytes [218,219]. Activation of these receptors by SPMs may attenuate neuroinflammation, which is known to be involved in pain/itch exacerbation [220,221,222,223]. Figure 4 schematically illustrates the hypothetical mechanisms of SPM action in resolving inflammation in AD. The beneficial effects of SPM in alleviating pain/itch also occur, among others, through the suppression of TRPV1 and TRPA1. For example, intrathecal capsaicin-induced spontaneous pain is blocked by RvE1 [224]. Capsaicin-induced pain is suppressed by MaR1, RvD2, and NPD1, but not RvD1, whereas allyl isothiocyanate-induced pain is inhibited by RvD1 and RvD2, but not RvE1, which suggests different modulation of inflammatory pain by SPM [219]. Moreover, in the IMQ-induced psoriasiform inflammation murine model, topical application and systemic administration of RvD3 resulted in reduced skin inflammation, acute pain and itch [208]. Mechanistically, RvD3 inhibited capsaicin-induced TRPV1 currents in DRG neurons via ALX/FPR2 with a concomitant decrease in calcitonin gene-related peptide (CGRP), which is also expressed by keratinocytes [225] and plays an apparent anti-inflammatory role in AD [226]. Taken together, these findings suggest that SPMs may alleviate AD-associated chronic pain/itch via effective control of skin inflammation by activation of SPM receptors on immune and non-immune cells.

## 6. Current AD Treatment

Recent substantial progress in deciphering the mechanisms of AD pathogenesis has brought to the forefront the immune dysregulation that we have discussed above and illustrated in Figure 1. This, in turn, has led to a real “translational revolution” [17], consisting of a significant expansion of the assortment of AD therapeutic agents, mainly including JAKi and monoclonal antibodies against key pathogenetic cytokines and their receptors [227,228,229,230]. The list of the most commonly used JAKi includes baricitinib, upadacitinib, and abrocitinib, which inhibit the signaling pathways of key cytokines involved in inflammatory diseases [231,232,233,234]. Among the monoclonal cytokine inhibitors, the most comprehensive clinical data are available for dupilumab, lebrikizumab and tralokinumab. Dupilumab acts against IL-4Rα, a common receptor to both IL-4 and IL-13. Lebrikizumab binds soluble IL-13 with a high affinity, preventing IL-4Rα/IL-13Rα1 heterodimerization (type 2 receptor). Tralokinumab binds IL-13 [61,62,235]. A comprehensive analysis of the clinical efficacy of current AD treatments has shown that high-dose upadacitinib, abrocitinib and low-dose upadacitinib are the most effective in addressing multiple patient-important outcomes. However, these JAKi are among the most harmful in terms of increasing adverse events. Dupilumab, lebrikizumab, and tralokinumab have intermediate efficacy and are among the safest. Low-dose baricitinib is one of the least effective [29].

Notably, although the therapeutic armamentarium for AD is rapidly expanding, the results of clinical trials have demonstrated that a significant number of patients fail to achieve clear or almost clear status (Investigator Global Assessment (IGA) score 0/1) with currently available treatments. For example, dupilumab achieves this score in <40% of patients when given as monotherapy or in combination with topical corticosteroids (TCSs) [17]. Less than 25% and <40% of tralokinumab-treated patients reach this score when used as monotherapy or in combination with TCSs [236]. JAK inhibitors demonstrate higher rates of achieving this score, with upadacitinib able to clear ~60% of patients and abrocitinib <50%, with slightly higher rates when used in combination with TCSs [17,29,237].

As discussed in the review, potential targets for alleviating the manifestations of AD may include agents that reduce itch. Indeed, preliminary data suggest that topical application of a cream containing N-palmitoylethanolamine (PEA), a TRVP1 antagonist, reduces itch and improves quality of life in patients with mild to moderate AD [238]. Results from the phase III study (NCT02965118) have not yet been published.

## 7. Application of Epidrugs, Dysbiosis Correctors and Inflammation-Resolving Agents for Alleviation of AD Manifestations

The lack of efficacy of JAKi and anti-cytokine antibodies in a significant proportion of patients with AD, as well as the unclear effect of anti-itch drugs, highlight the need for additional therapeutic approaches that could improve the outcome of AD treatment. Several potential options are schematically presented in Figure 5. The proposal starts with epidrugs because of the significant role of epigenetic disturbances in the pathogenesis of AD. Epidrugs, which are chemical factors that inhibit enzymes with epigenetic activity, are capable of restoring disrupted pathological changes in epigenetic mechanisms [239]. They, therefore, are possible candidates for future AD treatment based on their ability to control gene expression directly at the pre-transcriptional stage, thereby correcting gene dysregulation at its source [240]. Epidrugs target epigenetic marks, which are responsible for epigenetic alterations, such as DNMT and HDAC, or miRNAs. These drugs inhibit or activate disease-associated epigenetic proteins and lead to the improvement, treatment, or prevention of diseases [239,240].

Currently, epidrug research and clinical translation are mostly focused on cancer [241]. However, there is also research on other diseases, such as osteoarthritis [242], selected autoimmune diseases [243], and mental and neurodegenerative disorders [244]. Unfortunately, till the present time, clinical trials of epidrugs have led only to modest success alongside notable side effects, such as high toxicity and drug resistance. This is most probably due to the effects on off-target genes, presumably because many epigenetic regulators are not very specific in gene targeting [245]. Moreover, epidrugs may compromise gene regulation and the genomic stability of normal cells [246], altogether limiting their applicable therapeutic use. Various efforts are currently underway to overcome these obstacles and other issues arising from epigenome editing, including improvements in target specificity, enzymatic activity and drug delivery [245]. Another possibility is to combine genetic and epigenetic approaches, in which synthetic modules consisting of epigenetic regulators are coupled to a custom-designed genome targeting system (e.g., CRISPR-Cas9-based). These synthetic modules can provide precise targeting of the epigenetic regulator gene or locus, especially when used as an inhibitor [246,247,248]. This molecular engineering strategy has been explored to target miRNAs [249]. For example, the knockdown of miRNA-155 (which enhances CTLA-mediated T-cell activation) by CRISPR/Cas9 in macrophage cell lines showed a significant reduction in the development of rheumatoid arthritis-related symptoms [250]. In addition, the pan-HDAC inhibitor, belinostat, resolved skin barrier defects in AD by targeting the dysregulated miR-335:SOX6 axis [124]. In several AD murine models, selective HDAC6 inhibitor tubastatin A and selective SIRT1 inhibitor sirtinol attenuated symptoms associated with AD, accompanied by reduced expression of PGE2 and COX2, and serum levels of TSLP and chemokine CXCL13 [251]. HDAC6 knockout in mice prevented HDAC6-mediated passive cutaneous and systemic anaphylaxis [252]. Taken together, these observations suggest the potential application of epidrugs in AD.

Another potential therapeutic target is the treatment of AD-associated skin and gut dysbiosis. The results of some clinical studies [253,254,255,256] and most meta-analyses report clear beneficial effects of prebiotic/probiotic supplementation in children [257,258,259,260] and adults [260,261,262,263,264], although some meta-analyses found no difference between treatment and placebo groups [265,266]. The mechanisms by which prebiotic/probiotic supplements may benefit AD remain unclear. However, studies in other diseases associated with chronic inflammation suggest that the underlying mechanism is their ability to enhance the production of anti-inflammatory cytokines, such as IL-10 and suppress the production of pro-inflammatory cytokines, such as IL-1, IL-6, and TNFα [267,268,269,270,271,272,273]. This, in turn, suggests that the attenuation of AD resulting from the microbiota restoration occurs through the resolution of chronic inflammation.

PUFA supplementation and numerous preclinical studies of SPMs in AD models have revealed clear beneficial effects, supporting the validity of the idea of using agents that resolve chronic inflammation to treat AD. In this regard, it is noteworthy that SPMs facilitate the termination of the inflammatory response and initiate tissue repair and healing without being immunosuppressive [165,274,275]. This is in stark contrast to JAKi, anti-cytokine antibodies and corticosteroids known to suppress immune responses to pathogens. Furthermore, corticosteroids reduce SPM production and block the inhibitory effects of 17-HDHA and RvD1 on IgE production by B cells from asthma patients [276], thus potentially delaying inflammation resolution.

An additional and intriguing aspect of the potential therapeutic activity of SPMs is their ability to modulate HDACs, mainly SIRTs, confirming a link between epigenetics and chronic inflammation [277,278] and suggesting that combining pro-resolving compounds with epidrugs may enhance their efficacy. Consistent with this idea, RvE1 and lipoxin A4 (LXA4), acting via ChemR23 and ALX/FPR2 receptors, respectively, inhibited NF-κB activation and increased the expression of SIRT1, SIRT6 and SIRT7 in cultured human dental pulp fibroblasts in a synergistic manner [279]. In this study, co-administration of RvE1 and LXA4 markedly promoted the resolution of mouse pulpitis. In septic mice, RvD1 improved animal survival and attenuated lung inflammation by downregulating STAT3, NF-κB, ERK, and p38 expression through a mechanism partly dependent on SIRT1 [280]. In mice with cerebral ischemia/reperfusion, MaR1 attenuated mitochondrial damage and reduced TNFα and IL-1 production via triggering SIRT1 signaling, resulting in a reduction in infarction size and subsequent neurological defects [281]. In human macrophages and PBMCs, MaR1 reversed LPS-induced increased expression and secretion of TNFα, IL-1β and IL-6, concomitantly with increased expression of SIRT1, PGC-1α, and PPARγ [282]. Moreover, SPMs were shown to alleviate inflammation through the regulation of miRNAs. For example, in a murine model of collagen-induced arthritis, RvD1 decreased pannus formation and cartilage damage through the upregulation of miRNA-146a-5p [283]. In a murine model of systemic lupus erythematosus, RvD1 effectively ameliorated disease progression through up-regulating Tregs and down-regulating Th17 cells via miR-30e-5p [284].

However, SPMs are complex molecules that are susceptible to rapid enzymatic degradation, which limits their translational potential [163]. To overcome this problem, stable, small-molecule SPM mimetics and receptor agonists have been developed. They have shown encouraging therapeutic efficacy in various preclinical models, likely by reversing failed chronic inflammation (Reviewed in [176,177,285]). As an example, BML-111, a synthetic analog of LXA4 that is an ALF/FPR2s agonist, showed beneficial effects in animal models of pancreatitis [286], experimental autoimmune myocarditis [287,288], and chronic obstructive pulmonary disease (COPD) [289]. All of the above-mentioned beneficial effects of SPM mimetics and receptor agonists have been obtained in experimental model systems. Yet, the field of “resolution pharmacology” [290] is rapidly evolving, and the first clinical trial results of pro-inflammatory compounds have already been published, providing some insight into their translational potential [173]. For example, a pilot study of inhaled BML-111 and lipoxin A4 mimetic 5(S),6(R) LXA4 methyl ester demonstrated preliminary efficacy in the treatment of childhood asthma with acute moderate episodes [291]. Noteworthy, the treated study participants did not experience any clinical adverse events according to blood, urine, and stool tests, electrocardiogram, and liver and kidney function tests on day 7 of the intervention. In support, mice treated with BML-111 or 5(S),6(R)-LXA4 methyl ester showed no significant differences in pulmonary, renal, or liver function. Another stable LXR4, 15(R/S)-methyl-lipoxin A(4) cream demonstrated therapeutic efficacy in patients with childhood AD [214]. In a double-blind, placebo-controlled, randomized, parallel-group study, the application of benzolipoxin A4 methyl ester (BLXA4), a stable LXA4 mimetic, has effectively reduced gingival inflammation without any adverse safety signals [292]. Notably, this study shows that the topical application of BLXA4, in addition to its local effects, significantly increased circulating SPM levels.

Collectively, these observations suggest that stable SPM mimetics and receptor agonists could be applied as pharmaceuticals in disorders characterized by chronic inflammation, including AD. Given the critical role of failed resolution of chronic inflammation in the development of AD, we propose the combination of inflammation-resolving agents with epidrugs and prebiotics/probiotics as a new therapeutic approach for AD to effectively relieve the disease manifestations (Figure 5).

## 8. Concluding Remarks

This review provides extensive data supporting the concept that failed resolution of chronic inflammation is a key pathogenic mechanism underlying the development of AD. The establishment of chronic inflammation occurs under the synergistic effects of at least three fundamental inflammation-promoting inducers: (i) the reduced production and functional activity of SPMs, (ii) skin and gut dysbiosis, and (iii) disturbed epigenetic regulation of AD-associated pathogenic mechanisms. As such, the diminution of the activity of these inducers could attenuate AD manifestations. In accordance with this idea, we propose a multilevel treatment strategy: (1) the application of stable, small molecule SPM mimetics and receptor agonists; (2) the recovery of dysbiosis manifestation by supplementation of prebiotics/probiotics; and (3) the implementation of epidrugs that restore impaired epigenetic mechanisms. This treatment strategy can be used in conjunction with current AD therapy, which mainly includes JAKi and anti-cytokine monoclonal antibodies, resulting in the gradual mitigation of AD (Figure 5).

As discussed in the review, stable SPM mimetics and receptor agonists demonstrated beneficial effects in numerous preclinical AD and inflammation models. This may address the shortcomings of current AD immunotherapies. A critical difference between pro-resolving mediators and anti-inflammatory drugs is that although most anti-inflammatory agents can cause immunosuppression, the resolution of inflammation occurs through active endogenous reprogramming of the immune response, allowing inflammation to cease without causing immunosuppressive effects. Recently, it has been found that an agonistic antibody against the ChemR23 receptor is capable of accelerating the resolution of acute inflammation, stimulating macrophage efferocytosis and reducing neutrophil apoptosis at the site of inflammation [293]; these impressive results allow this agonistic antibody to be included in the list of potential pro-resolving agents.

However, the implementation of the proposed combinatory therapeutic strategy requires overcoming critical research challenges. One of the major ones is the search for the optimal animal model of AD. Currently, murine models of AD are classified into three groups: (i) inbred models, (ii) genetically modified mice in which the genes of interest are overexpressed or deleted in a specific cell type, and (iii) models induced by local application of exogenous agents. However, each animal model represents only limited aspects of human AD, and a significant translational gap remains between mouse AD models and human AD [294,295,296]. Therefore, the selection of the most appropriate model or any new model to test our idea of combinatorial treatment of AD is a challenge and requires further research. What are the dominant factors driving epi-genetic dysregulation, dysbiosis and chronic inflammation in AD pathogenesis? Are the relationships between these pathogenic events and AD causal, or do they rather represent independent pathological processes? In this regard, identifying and characterizing the epigenetic, dysbiosis, and chronic inflammation profiles of AD patients could provide a unique opportunity to subtype patients, thereby gaining further insight into the disease. Furthermore, in order to harness the benefits of the proposed multilevel treatment strategy, the crucial next step is to conduct a scientific search for highly specific targets (receptors, cells, signaling pathways, and microbes). However, there is currently a wide variety of possible targets, making the development of such a complex therapeutic strategy a highly challenging task. Once we overcome this challenge, it will open the door to developing personalized treatments for AD patients to significantly reduce manifestations.

## Figures and Tables

**Figure 1 cells-13-01899-f001:**
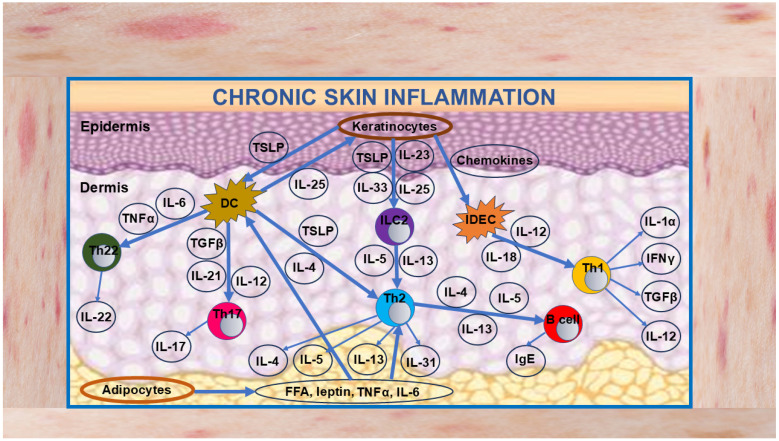
Schematic diagram illustrating the development of skin chronic inflammation in AD. Disruption of the epidermal barrier (mainly caused by mechanical scratch) activates and damages keratinocytes, which is accompanied by secretion of a wide range of pro-inflammatory cytokines and chemokines. Cytokines TSLP, IL-25, and IL-33, classified as alarmins, activate ILC2 cells. In response, ILC2 cells produce IL-5 and IL-13, which activate Th2 cells, leading to increased production of IL-4 and IL-13 that activate B cells to produce IgE. IL-5, produced by Th2 cells, chemoattracts eosinophils (not depicted in the diagram), presumably contributing to AD pathogenesis. Secreted by keratinocytes, TSLP activates dendritic cells (DCs), resulting in enhanced production of TNFα and IL-6, which, in turn, activate Th-22 cells. DCs secrete IL-25, which activates keratinocytes, whereas secreted by DCs IL-12, IL-21, and TGFβ activate Th-17 cells. In turn, IL-22 from Th22 cells and IL-17 from Th17 cells cause thickening of epidermis. Activation of inflammatory dendritic epidermal cells (IDEC) by chemokines TARC, CCL17, CTACK, and MDS leads to enhanced secretion of IL-12 and IL-18, which activate Th1 cells to produce IL-1α, IL-12, IFNγ, and TGFβ capable of keratinocyte stimulation, thereby worsening skin inflammation. IL-31 stimulates the differentiation of keratinocytes, whereas IL-25 and IL-33 promote itch (pruritus) and skin barrier disruption. Adipokines secrete pro-inflammatory molecules, such as free fatty acids (FFAs), leptin, TNFα, IL-6 and some others capable of activating DCs and Th2 cells. All these events create a vicious cycle that governs and maintains chronic skin inflammation in AD. Further explanations are given in the text. Abbreviations: CCL17, CC motif chemokine ligand 17; CTACK, cutaneous T-cell attracting chemokine; DC, dendritic cell; FFAs, free fatty acids; IDEC, inflammatory dendritic epidermal cell; ILC2, group 2 innate lymphoid cell, MDC, macrophage-derived chemokine; TGF, transforming growth factor; TARC, thymus- and activation-regulated chemokine; Th, T helper cell; TNF, tumor necrosis factor; TSLP, thymic stromal lymphopoietin.

**Figure 2 cells-13-01899-f002:**
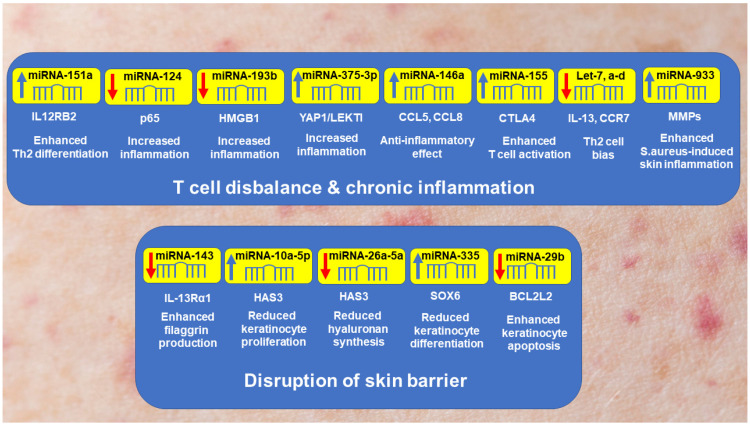
Schematic presentation of the involvement of miRNAs in the pathogenesis of AD. The upper panel of the figure indicates that down- or up-regulation of the expression of miRNAs leads to T-cell imbalance as a prerequisite for the development and maintenance of chronic inflammation. The lower panel shows skin barrier disruption that exacerbates AD manifestations. Down-arrows and up-arrows show decreased and increased expression of miRNA, respectively. Further explanations are given in the text. Abbreviations: Bcl2L2, Bcl-2-like protein 2; CTLA4, cytotoxic T-lymphocyte associated protein 4; HAS3, hyaluronan synthase 3; HMGB1, high mobility group box 1 protein; IL12RB2, IL-12 receptor subunit beta 2; IL-13Rα1, IL-13 receptor subunit alpha 1; LEKTI, lympho-epithelial Kazal type inhibitor; MMPs, matrix metalloproteinases; SOX6, transcription factor SOX-6; YAP1, Yes-associated protein.

**Figure 3 cells-13-01899-f003:**
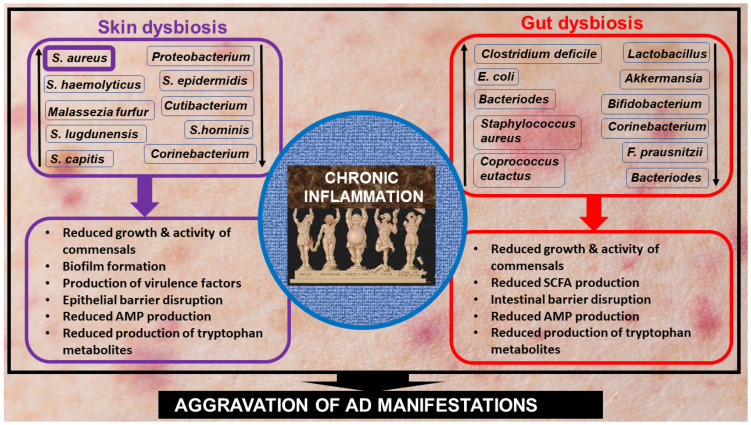
Schematic presentation of the skin-gut axis involvement in chronic inflammation development associated with AD pathogenesis. Skin dysbiosis (left-hand side of the diagram) in AD is characterized by a clear predominance of *Staphylococcus aureus* (*S. aureus*) along with an increase in the prevalence of some other pathogenic microorganisms (pathobionts) and a decrease in the prevalence of commensals. This leads to the formation of biofilm that weakens the host’s immune response as well as the production of several virulence factors, such as toxins and proteases that damage the epithelial barrier and increase histamine release and IgE levels. Reduced production of antimicrobial peptides (AMPs) (i.e., defensins and Cathelicidin LL-37) and tryptophan metabolites (i.e., indole-3-aldehyde) leads to decreased skin protective activity and increased secretion of TSLP by keratinocytes, thereby deteriorating the epidermal barrier of the skin. Gut dysbiosis (right-hand side of the diagram) is characterized by a decrease in the prevalence of microorganisms that produce short-chain fatty acids (SCFAs) (i.e., butyrate and propionate), which leads to disturbed barrier integrity, lipopolysaccharide (LPS) leakage, increased and reduced production of pro- and anti-inflammatory cytokines, respectively. Similar to skin dysbiosis, gut dysbiosis is also characterized by reduced production of AMPs and tryptophan metabolites. Overall, these events create a skin-gut axis that contributes to AD-associated chronic inflammation and worsening of AD manifestations. Further explanations are given in the text. Abbreviations: AMP, antimicrobial peptides; LPS, lipopolysaccharide; SCFAs, short-chain fatty acids; *S. aureus*, *Staphylococcus aureus*; TSLP, thymic stromal lymphopoietin.

**Figure 4 cells-13-01899-f004:**
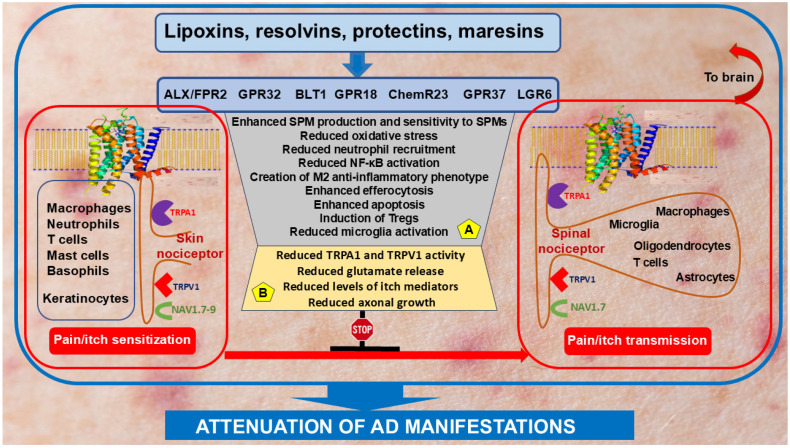
Hypothetical mechanisms underlying the role of SPMs in inflammation resolution and attenuation of pain and itch in AD. SPMs (upper line of the diagram) transmit signals through an array of receptors expressed in various immune cells, keratinocytes and skin nociceptors, which also express transient receptor potential (TRP) ion channels, mainly the vanilloid (TRPV) and ankyrin 1 (TRPA1) subtypes, and voltage-gated sodium ion channels (Nav 1.4, Nav1.7-9) (left-hand side of the diagram). These pain/itch-signaling receptors are also expressed in spinal nociceptors, which also express SPM receptors on indicated immune and neuronal cells (right-hand side of the diagram). The inflammation-resolving effects provided by SPMs through binding to their receptors on immune (**A**) and neuronal (**B**) cells ultimately result in suppression of pain/itch signaling, thereby attenuating AD manifestations. Further explanations are given in the text. Abbreviations: M2, anti-inflammatory subtype of macrophages; NF-κB, nuclear factor kappa B.

**Figure 5 cells-13-01899-f005:**
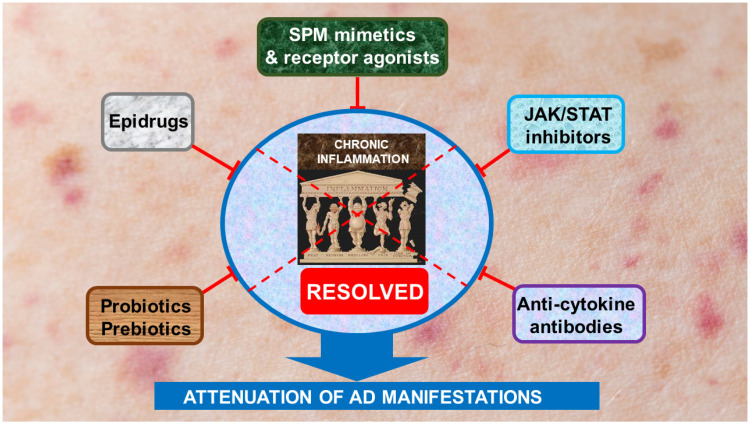
A proposed therapeutic strategy to resolve failed chronic inflammation in AD. We suggest that failed chronic inflammation is a key element in AD pathogenesis. This occurs under the synergistic actions of at least three processes: (1) the reduced production and functional activity of SPMs, (2) skin and gut dysbiosis; and (3) epigenetic disturbances. In accordance, diminution of the activities of these chronic inflammation inducers could reverse AD manifestations. To achieve this overall aim, we propose a multilevel approach: (1) the application of stable, small-molecule SPM mimetics and receptor agonists; (2) the convalescence of dysbiosis manifestation by supplementation of probiotics/prebiotics; and (3) the correction of epigenetic disturbances by epidrugs. This treatment strategy can be used in conjunction with current AD therapy, which mainly includes JAKi and anti-cytokine monoclonal antibodies. This combined approach, in our view, would lead to the resolution of chronic inflammation, ultimately leading to the gradual amelioration of AD manifestations. Further explanations are provided in the text.

## Data Availability

No data were used for the research described in the article.

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
