# Peer review of "Resolution of Chronic Inflammation, Restoration of Epigenetic Disturbances and Correction of Dysbiosis as an Adjunctive Approach to the Treatment of Atopic Dermatitis"

_cells, 2024, doi:10.3390/cells13221899_

Round 1
Reviewer 1 Report
Comments and Suggestions for Authors
The presented manuscript entitled “Resolution of chronic inflammation, restoration of epigenetic disturbances and correction of dysbiosis as an adjunctive approach to the treatment of atopic dermatitis” is a very nice review - very well prepared, showing a wide range of literature, although sometimes it is very basic knowledge. On the Figure 1 the interaction between keratinocytes and DC is not described. Moreover, I would suggest adding a description of the role of lipid peroxidation/metabolism products in pro-inflammatory signaling, especially since the authors refer to PUFA supplementation in AD therapy.
Author Response
Reviewer #1
The presented manuscript entitled “Resolution of chronic inflammation, restoration of epigenetic disturbances and correction of dysbiosis as an adjunctive approach to the treatment of atopic dermatitis” is a very nice review - very well prepared, showing a wide range of literature, although sometimes it is very basic knowledge.
We are very thankful to our reviewer for his/her positive assessment of our manuscript and also greatly appreciate the constructive and helpful suggestions. We set out to fully address the raised concerns.
- On the Figure 1 the interaction between keratinocytes and DC is not described.
Authors’ response: We agree with this comment and to meet it, we have included a brief description of the interaction between keratinocytes and DCs in Figure 1 and the legend. We have added TSLP and IL-25 as the main communicators between keratinocytes and DCs with reference to the PMID: 35967329 and PMID: 38590314. We have also related to this issue in the figure legend in the revised manuscript (page 4, lines 161-170.
- Moreover, I would suggest adding a description of the role of lipid peroxidation/metabolism products in pro-inflammatory signaling, especially since the authors refer to PUFA supplementation in AD therapy.
Authors’ response: We thank our reviewer for this comment, which raises the important and controversial issue on the link between PUFA oxidation and their biological activity. Indeed, it is well established that lipid oxidation reactions accompany oxidative stress. This leads to the formation of highly bioactive lipid peroxidation products, such as 4-hydroxynonenal (4-HNE) and malondialdehyde (MDA), participating in PUFA oxidation (PMID: 28212725). In the present review, we mentioned that supplementation of n-3 PUFAs alleviates AD manifestations in both preclinical and clinical studies (lines 63, 64). However, although omega-3 PUFAs are suggested to possess anti-oxidative properties, these molecules are highly oxidizable due to multiple double bonds and may increase oxidative stress (PMID: 23746276; PMID: 34915303). Moreover, studies (PMID: 33670710) showed that omega-3 PUFA supplementation is associated with increased lipid peroxidation and lipid peroxidation products are known to induce inflammation and other immune alterations that include increased cytokine secretion and activation of inflammatory transcription factors (PMID: 28441057; PMID: 28819546). Overall, these results indicate the existence of very complex relationships between oxidation of PUFAs and their biological activity (PMID: 33670710; PMID: 34660660; PMID: 39290706). We discussed this issue in the revised manuscript (page 11, lines 497-505).
Reviewer 2 Report
Comments and Suggestions for Authors
Title
Could you include the study type in the title for clarity and consistency?
Narrative Structure
- Lines 27-33: Including this information would offer a more holistic view of the multifactorial nature of AD development.
- Lines 49-52: Balancing the discussion across these different mechanisms would offer a more comprehensive analysis.
- Lines 58-67: Expanding on the link between these sections would improve the flow and clarify how genetic predispositions impact the failure of inflammation resolution mechanisms.
- Lines 112-125: Dividing this section into smaller, focused paragraphs could enhance readability, especially for those less familiar with the complex immunopathological processes.
- Incorporate recent literature on pain-related mechanisms in osteoarthritis.
PMID: 25557054
PMID: 21899891
PMID: 35627729
Description of Methods
- Lines 58-67: Providing specific details on the assays, models, or experimental protocols used in these observations would enhance the transparency and reproducibility of the findings.
- Lines 182-229: The discussion of epigenetic regulation is thorough, but the methods for assessing these epigenetic modifications (e.g., DNA methylation, histone modifications) need elaboration. Were techniques like bisulfite sequencing or ChIP-seq employed? Clarifying this would add rigor to the methodological section.
Tables and Figures
- Lines 142-155 (Figure 1): The diagram explaining the development of skin inflammation in AD is valuable but could be improved by refining the color-coding scheme for immune cells, cytokines, and chemokines. A more detailed legend would further assist readers in interpreting the figure accurately.
- Figure 4 (Lines 552-561): While the diagram effectively summarizes the role of SPMs in resolving inflammation, incorporating a flow chart to depict the sequence of these interactions would improve comprehension, especially regarding the complex interactions between immune cells and SPM receptors.
Author Response
Reviewer #2
Lines 58-67: Expanding on the link between these sections would improve the flow and clarify how genetic predispositions impact the failure of inflammation resolution mechanisms.
Authors’ response: We are grateful to our reviewer for this interesting comment, which raises an important, but still understudied question concerning the role of the genetic factors in the process of inflammation resolution. Although the genetic aspects of inflammation have been extensively investigated (e.g., PMID: 15385745; PMID: 19570714; PMID: 27303926), the issue of genetic factors involvement into the process of the inflammation resolution remains poorly understood in resolution biology. Only very limited published data available on this subject. For example, study of macrophages, playing a key role in inflammation resolution through efferocytosis of neutrophils, revealed that long noncoding RNA (lncRNA), lncFAO, contributes to inflammation resolution and tissue repair in mice by promoting fatty acid oxidation in macrophages (PMID: 32513690). This information has been included in the section “Epigenetic profile of AD” in the revised manuscript (page 8, lines 344-345).
Two other studies on this topic implemented Mendelian randomization analysis. One of them suggested that a higher level of omega-3 fatty acids is associated with a decreased chance of developing AD, indicting a protective effect of the omega-3 fatty acids, including DHA, in AD (PMID: 36908902). A protective effect was mainly driven by one SNP (rs174546), which is mapped to the fatty acid desaturase (FADS) gene cluster, highlighting FADS significance in the fatty acid synthesis pathway in the development of AD (PMID: 36908902). In another study (PMID: 38235898), DHA levels showed a negative association with the levels of pro-inflammatory cytokine TNFSF14. Moreover, a positive correlation was found between TNFSF14 levels and AD risk. Taken together, these sparse findings point to a possible involvement of the genetic factors in the molecular mechanisms responsible for a failure to resolve inflammation in general and in AD, in particular. We have discussed this issue in the revised manuscript (p. 11, lines 482-491)
- Lines 112-125: Dividing this section into smaller, focused paragraphs could enhance readability, especially for those less familiar with the complex immunopathological processes.
Authors’ response: We thank the reviewer for this comment and divided this section as recommended in the revised manuscript (page 3, lines 111-138).
- Incorporate recent literature on pain-related mechanisms in osteoarthritis.
PMID: 25557054
PMID: 21899891
PMID: 35627729
Authors’ response: We are concerned that there has been a misunderstanding or error. All three of the recommended studies address several aspects of osteoarthritis research. For example, PMID: 25557054 focuses on “grip and pinch strength differences in women with thumb carpometacarpal osteoarthritis.” These questions are not related to atopic dermatitis, the subject of this review.
Description of Methods
- Lines 58-67: Providing specific details on the assays, models, or experimental protocols used in these observations would enhance the transparency and reproducibility of the findings.
Authors’ response: Since this is only an introduction to the topic, we believe that a detailed description of the assays, models, or experimental protocols should be presented in a section where functional activity of SPMs is described, namely in the section “Failed resolution of chronic inflammation as a key mechanism of AD pathogenesis”, which was indeed done (page 10). Nevertheless, following the reviewer’s comment, we added some specific details on the assays, models, or experimental protocols used in the respective studies (please see lines 507, 508, 515, 518, page 11; lines 534, 561, page 12; line 575, page 13).
Lines 182-229: The discussion of epigenetic regulation is thorough, but the methods for assessing these epigenetic modifications (e.g., DNA methylation, histone modifications) need elaboration. Were techniques like bisulfite sequencing or ChIP-seq employed? Clarifying this would add rigor to the methodological section.
Authors’ response: We thank the reviewer for this important comment. Accordingly, in the revised manuscript, we have added a description of the methods used to assess epigenetic modifications (please see lines 234-239, page 5 and 6; lines 249-252, page 6; lines 344-346, page 8).
Tables and Figures
- Lines 142-155 (Figure 1): The diagram explaining the development of skin inflammation in AD is valuable but could be improved by refining the color-coding scheme for immune cells, cytokines, and chemokines. A more detailed legend would further assist readers in interpreting the figure accurately.
Authors’ response: We are grateful to the reviewer for this comment, which helped us improve the figure and the legend. Accordingly, we have changed (enhanced) the color of some immune cells presentation, which makes them more visible and contrasting. However, we think that adding different colors to cytokines and chemokines would make the diagram motley, and difficult for the reader to perceive. Furthermore, following the comment of Reviewer #1, we have clarified the relationships between keratinocytes and dendritic cells. In addition, following the corresponding comment of Reviewer #3, we have added adipocytes and secreted molecules to the figure. Overall, this led to the substantial modification and complication of the figure and, accordingly, the accompanying legend.
- Figure 4 (Lines 552-561): While the diagram effectively summarizes the role of SPMs in resolving inflammation, incorporating a flow chart to depict the sequence of these interactions would improve comprehension, especially regarding the complex interactions between immune cells and SPM receptors.
Authors’ response: We thank the reviewer for this comment. Following a reviewer's suggestion, we prepared a modified version of this figure in the revised manuscript. We have changed the order of the processes presented in the center of the figure (panel A), which now actually reflects the sequence of events occurring during inflammation resolution (PMID: 23197111; PMID: 26037968), which is now also described in the revised manuscript (please see page 10). In addition, we have also changed the order of the processes presented in the center of the figure (panel B), which is now presents the sequence of events occurring during beneficial effects of SPMs on itch transmission in AD (please see pages 12 and 13).
Reviewer 3 Report
Comments and Suggestions for Authors
This review describes atopic dermatitis and underlying mechanisms. Key factors include epigenetic dysregulation and microbiota dysbiosis, leading to chronic inflammation. Reduced levels of specialized pro-resolving mediators in atopic dermatitis patients suggest PUFA supplementation, SPM mimetics, prebiotics, probiotics, and epidrugs as potential therapies to alleviate atopic dermatitis.
This is a well-written and comprehensive review on atopic dermatitis.
only several points should be improved.
1 fig1 adipocytes should be described.
2 figure2 figures are unclear and misunderstanding. It seems that they represent mRNAs, but might cause misunderstanding. These should be deleted.
Author Response
Reviewer #3
This review describes atopic dermatitis and underlying mechanisms. Key factors include epigenetic dysregulation and microbiota dysbiosis, leading to chronic inflammation. Reduced levels of specialized pro-resolving mediators in atopic dermatitis patients suggest PUFA supplementation, SPM mimetics, prebiotics, probiotics, and epidrugs as potential therapies to alleviate atopic dermatitis.
This is a well-written and comprehensive review on atopic dermatitis. Only several points should be improved.
We are very grateful to the reviewer for the positive evaluation of our manuscript and appreciate the provided suggestions, and pleased to meet them fully.
fig1 adipocytes should be described
Authors’ response: We agree with this comment and have modified Figure 1 and the corresponding legend in the revised manuscript accordingly (page 4, lines 167-170).
figure2 figures are unclear and misunderstanding. It seems that they represent mRNAs, but might cause misunderstanding. These should be deleted.
Authors’ response: We respectfully disagree with the reviewer on this point. This figure schematically summarizes the information discussed in this review on pages 6-8 regarding mRNAs involved in AD pathogenesis and their underlying mechanisms of action. For example, it has been shown that AD pathogenesis involves the infiltration of activated memory CD4+ T cells into the skin. The figure 2, among other findings, shows that down- or up-regulation of miRNA expression leads to T-cell imbalance as a prerequisite for the development and maintenance of chronic inflammation. The latter is a major factor to onset of allergic inflammation in a patients with AD. We, therefore, believe that the information presented in figure 2 is important and helpful for better understanding of AD pathogenesis.